# Regional disparities in antenatal care utilization in Indonesia

**Agung Dwi Laksono**[1]**, Rukmini Rukmini**[1]**, Ratna Dwi Wulandari**[ID][2]*

**1** National Institute of Health Research and Development, The Ministry of Health, Jakarta, The Republic of Indonesia, **2** Faculty of Public Health, Universitas Airlangga, Surabaya, Indonesia

* ratna-d-w@fkm.unair.ac.id

## Abstract

### Introduction

The main strategy for decreasing maternal morbidity and mortality has been antenatal care (ANC). ANC aims to monitor and maintain the health and safety of the mother and the fetus, detect all complications of pregnancy and take the necessary actions, respond to complaints, prepare for birth, and promote a healthy lifestyle. This study aims to analyze interregional disparities in $\geq 4$ ANC visits during pregnancy in Indonesia.

### Methods

Data was acquired from the 2017 Indonesian Demographic and Health Survey (IDHS). The unit of analysis was women aged 15–49 years old, and a sample of 15,351 women was obtained. In addition to ANC as the dependent variable, the other variables analyzed in this study were a place of residence, age, husband/partner, education, parity, wealth status, and health insurance. For the final analysis, binary logistic regression was used to determine disparity.

### Results

With the Papua region as a reference, all regions showed a gap except for the Maluku region, which was not significantly different in the use of ANC compared to the Papua region. Women in the Nusa Tenggara have 4.365 times the chance of making $\geq 4$ ANC visits compared to those in the Papua region (95% CI 3.229–5.899). Women in Java-Bali have 3.607 times the chance of making $\geq 4$ ANC visits compared to women in the Papua region (95% CI 2.741–4.746). Women in Sumatra have 1.370 times the chance of making $\geq 4$ ANC visits compared to women in the Papua region (95% CI 1.066–1.761). Women in Kalimantan have 2.232 times the chance of making $\geq 4$ ANC visits compared to women in the Papua region (1.664–2.994). Women in Sulawesi have 1.980 times more chance of making $\geq 4$ ANC visits compared to women in the Papua region (1.523–2.574). In addition to the region category, other variables that contributed to the predictor were age, husband/partner, education, parity, wealth and insurance.

**Data Availability Statement:** Data cannot be shared publicly because of ethical restrictions prohibit public sharing of a data set. Data is available from the https://dhsprogram.com/data/new-user-registration.cfm by submitting an

application to the ICF via the website. Other researchers will be able to access the data set in the same way as the authors, and the authors do not have special access rights that others do not have. Interested researchers can replicate the findings in this study as a whole by directly obtaining data from IDHS by following the protocol in the manuscript method section. The IDHS data set that I use is a data set of "women of childbearing age" in Indonesia.

**Funding:** The author(s) received no specific funding for this work.

**Competing interests:** The authors have declared that no competing interests exist.

## Conclusion

There were disparities in ANC utilization between the various regions of Indonesia. The structured policy is needed to reach regions that have low coverage of ≥4 ANC. Policy-makers need to use the results of this study to take the necessary policies. Policies that focus on service equality to reduce disparities.

## Introduction

Indonesia has entered the final year of the 2015–2019 National Medium-Term Development Plan. In the 2015–2019 National Medium-Term Development Plan, 4 main health targets were established, which must be achieved by 2019: 1) Improve the health and nutritional status of the community; 2) Improve the control of communicable and noncommunicable diseases; 3) Increase the equity and quality of health services; and 4) Increase financial protection, availability, distribution, quality of medicines and health resources [1].

In health development, the target of increasing equal distribution and quality of health services is determined by three indicators, namely, the number of subdistricts that have at least one accredited Puskesmas (Health Center), which is 5,600; the number of regencies/cities that have at least one nationally accredited hospital, which is 481; and the percentage of regencies/cities that have up to 80% completed basic immunizations in infants, which is as much as 95%. Based on the Ministry of Health's report, this target has been achieved; in 2018, the target number of subdistricts that had at least one accredited Health Center of the 4,900 subdistricts has been as many as 5,385 subdistricts (109.9%) or approximately 7,518 Health Centers. This achievement exceeded the established target because several regencies/cities used the Regional Revenue and Expenditure Budget purely for the accreditation process and did not use resources from the Non-Physical Allocation Fund. In 2018, the number of regencies/cities that had at least one nationally accredited hospital was 440 (101.4%) of the target of 434. The immunization target was not achieved; the 2018 data shows that complete basic immunization coverage for children aged 12–23 months in Indonesia was 57.9%, incomplete coverage was 32.9% and not immunized was 9.2% [2].

With regard to the target of improving the community's health and nutrition status, several achievement targets have been set, namely, a maternal mortality rate (MMR) of 306/100,000 live births, an infant mortality rate (IMR) of 24/1,000 live births, a prevalence of malnutrition in children under five of 17/100,000, and a prevalence of stunting in children under two years of 28/100,000. The MMR is currently reported to have decreased by 346 deaths to 305 maternal deaths per 100,000 live births but has not reached the MDG target in 2015 of 102/100,000 live births [3]. On the other hand, Indonesia must strive to be higher on the SDG's target by reducing the MMR to below 70/100,000 live births, reducing neonatal mortality to 12/1000 live births and reducing the toddler death rate to 25/1000 live births [4]. The MMR in Indonesia is the highest compared to other ASEAN countries and is 9 times that of Malaysia, 5 times that of Vietnam and almost 2 times that of Cambodia. Based on the WHO reports, the estimated MMR in developed countries is 12/100,000 live births, while in developing countries, it is 239/100,000 live births [5, 6].

The results of research in Indonesia that used 2013 data showed disparities in maternal deaths among districts/cities in Indonesia, with the highest risk of maternal deaths occurring in Eastern Indonesia. The risk factors that most influenced maternal mortality were population density with OR 0.283 (95% CI 0.185–0.430) and delivery by health workers with OR 1.745

(95% CI 1.081–2.815). The risk of maternal death is high in districts/cities with low coverage of fourth pregnancy visits, low coverage of delivery by health workers, low coverage of postpartum visits, high average number of children, low average length of schooling for women of childbearing age, and high poverty [7].

The main strategy is to decrease maternal morbidity and mortality with antenatal care (ANC). ANC aims to monitor and maintain the health and safety of the mother and the fetus, detect all complications of pregnancy and take the necessary actions, respond to complaints, prepare for birth, and promote a healthy lifestyle. ANC visits are very important to detect and prevent unwanted occurrences that arise during pregnancy [8]. In developing countries, there has been an increase in the utilization of maternal health services, but it still varies among population groups. Disparities can occur due to geographical, demographic, socioeconomic, and cultural differences. Gaps that occur result in decreased access to services, service quality, and service affordability [9, 10].

Several research results show that interregional disparities occur in several countries. In Nigeria, interregional disparities occur at the utilization of intermittent preventive treatment for malaria in pregnancy [11]. Interregional disparities occur in Bangladesh in maternal healthcare [12]. ANC service in Ethiopia was reported to experience interregional disparities over a period of eighteen years [13]. In line with this phenomenon, in Liberia, there have also been reported interregional disparities in the utilization of ANC services [14].

In 2018, there was an increase in the proportion of ANC visits for women aged 10–54 years, i.e., first visit by 96.1% compared to 95.2% in 2013, while ANC fourth visits in 2018 amounted to 74.1% compared to 70.0% in 2013; the coverage of ANC fourth visits is still below the target that was established in the 2017 Strategic Plan, which is 76.0% [15]. However, the quality of services to ensure early diagnosis and appropriate care for pregnant women still needs to be improved. Midwives are spearheading pregnancy checks by identifying complications or symptoms of complications, assisting in labor and conducting childbirth examinations. If there are signs of complications that cannot be treated, the midwife must make a referral to a health facility that provides Basic Emergency Neonatal Obstetric Services to obtain further treatment [16]. Data from the Ministry of Health in 2018 stated that the majority (62.7%) of deliveries were assisted by midwives and were carried out in independent midwife practices (29%), although there were still many carried out at home (16%) [15].

This study was conducted to analyze interregional disparities in the utilization of ≥4 ANC visits during pregnancy in women aged 15–49 years who gave birth in the last five years in Indonesia. This study is important because it can provide clear directions for the Ministry of Health to complete regional priority data in an effort to reduce maternal mortality.

## Methods

### Data source

This study analyzed data from the 2017 Indonesian Demographic Data Survey (IDHS). The IDHS was part of the International Demographic and Health Survey (DHS) program conducted by the Inner City Fund (ICF).

The 2017 IDHS sampling design was designed to present national and provincial level estimates. The 2017 IDHS sample includes 1,970 census blocks covering urban and rural areas. The census blocks were expected to obtain a household sample of 49,250 respondents. From all household samples, it was expected that 59,100 female respondents of childbearing age (aged 15–49 years) could be obtained. The 2017 IDHS sample framework uses the master census block sample from the 2010 Population Census. The household selection sample

framework uses the list of ordinary households that have been updated from the selected census block [17]. Women 15–49 years of age who had given birth in the last 5 years were the unit of analysis in this study. A sample of 15,351 women was obtained.

The sampling design used in the 2017 IDHS is a two-stage stratified sampling method. Stage 1 involved selecting a number of systematic census blocks in a systematic proportional to size (PPS) measure with the size of the households as a result of the 2010 Population Census listing. Stage 2 consisted of systematically selecting 25 ordinary households in each census block from the result of updating the households in each of the census blocks [17].

## Procedure

Ethical clearance was obtained in the 2017 IDHS from the National Ethics Committee. The respondents' identities have all been deleted from the dataset. Respondents provided written approval for their involvement in the study. Researchers have obtained permission to use the data for the purposes of this study through the following website: https://dhsprogram.com/data/new-user-registration.cfm.

## Data analysis

The Ministry of Health of the Republic of Indonesia recommends that the ANC during pregnancy be performed at least 4 times, namely, 1 time in the first trimester, 1 time in the second trimester, and 2 times in the third trimester [17]. The operational definition of ANC Utilization used in this study was the respondent's acknowledgment of the amount of ANC utilization during pregnancy. The ANC utilization was divided into 2 criteria, namely <4 and ≥4. The division of regions was grouped by the largest island. Divided into 7 regions, namely Sumatera, Java-Bali, Nusa Tenggara, Kalimantan, Sulawesi, Maluku Islands, and Papua [18].

Other variables analyzed as independent variables are the place of residence, age, husband/partner, education level, parity, wealth status, and health insurance. Including socio-demographic variables in the analysis of this study was important, because it will provide more specific targets for policymakers.

The place of residence was divided into urban or rural residential areas. The urban-rural designation follows the criteria issued by the Central Statistics Agency. Age was the respondent's acknowledgment of the last birthday she has passed. Ages were grouped in five years, so they form seven groups. The husband/partner was a respondent's acknowledgment of husband/partner ownership during pregnancy. Education level was the respondent's acknowledgment of the last educational level that was passed. Education level was divided into 4 criteria, namely no education, primary, secondary, and higher. Parity was the respondent's acknowledgment of the number of live babies ever born. Parity was divided into 3 criteria, namely <2, 2–4, and >4. Wealth status was based on the wealth quintile owned by a household. Households were scored based on the number and type of items they have, from televisions to bicycles or cars, and housing characteristics, such as drinking water sources, toilet facilities, and main building materials for the floor of the house. This score was calculated using principal component analysis. National wealth quintiles were arranged based on household scores for each person in the household and then divided by the distribution into the same five categories, with each accounting for 20% of the population [17]. Health insurance was the respondent's acknowledgment of health insurance ownership.

The variable inclusion criteria taken from IDHS were all women 15–49 years of age who had given birth in the last 5 years. The exclusion criteria were the variables were not available or the variables were incomplete.

Because all of the variables are categorical, the chi-square test was used to select variables related to the frequency of ANC utilization during pregnancy. Because of the nature of the dependent variable, binary logistic regression was used for the final test to determine disparity.

## Results

Fig 1 is a description of the distribution of ≥4 ANC visits in 34 provinces in Indonesia. The eastern part of Indonesia (Maluku and Papua regions) has the lowest distribution of ≥4 ANC visits. The western most area (part of the Sumatra region) has a distribution of ≥4 ANC visits at one level above. The distribution of ≥4 ANC visits is best centered on the central region of the Java-Bali region.

### Socio-demographic characteristics

The statistical description of female respondents aged 15–49 years who gave birth in the last five years in Indonesia is presented in Table 1. Table 1 shows that there are statistically significant differences between regions. Each region was dominated by the use of ANC, which had ≥4 visits.

Table 1 indicates that the Java-Bali and Kalimantan regions are more dominated by urban areas, while the remaining regions are dominated by rural areas. In all regions, it was also seen that the dominant age categories of women were 25–29 years and 30–34 years. Table 1 shows that all regions are dominated by women who have a husband/partner, have a secondary education level, and have 2–4 parity.

Table 1 shows that almost all regions are dominated by women who have the wealth status of 'poorer' or 'poorest', except in the Java-Bali region, which is dominated by women with the 'richest' wealth status. Most women aged 15–49 years who had delivered a baby in their last five years in Indonesia were covered by health insurance in all regions.

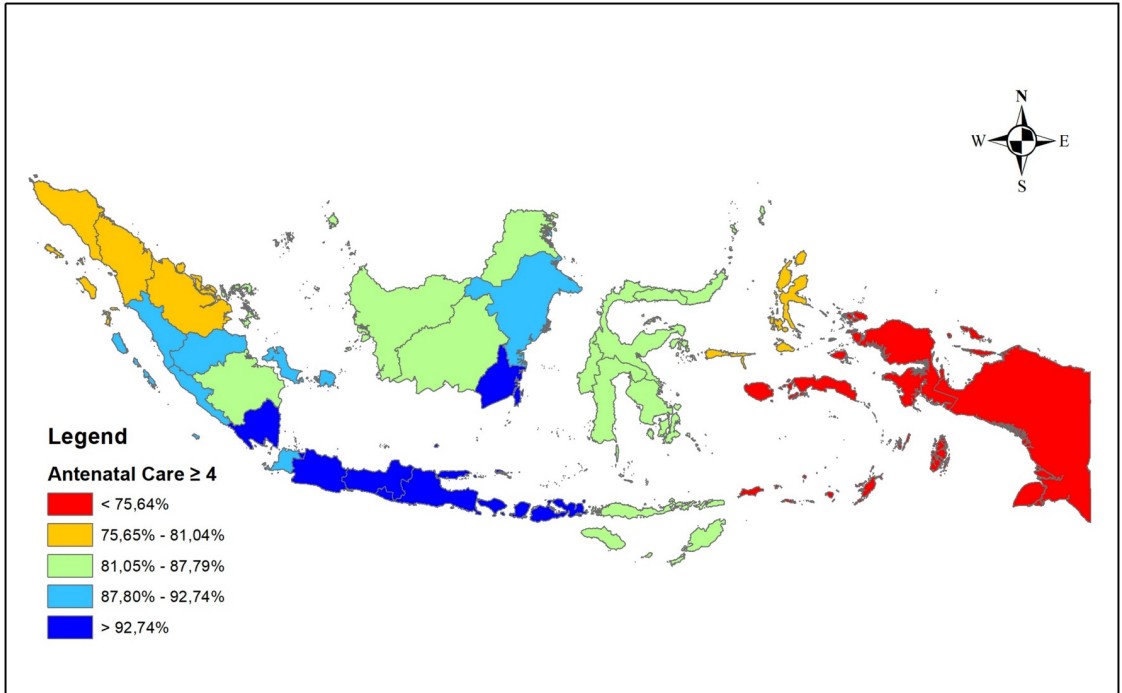

**Fig 1. Distribution of ≥4 ANC visits by province in Indonesia.**

**Table 1. Socio-demographic of respondents (n = 15,351).**

| Variables | Sumatera | Java-Bali | Nusa Tenggara | Kalimantan | Sulawesi | Maluku Islands | Papua | All | P |
|---|---|---|---|---|---|---|---|---|---|
| | | | | **Region** | | | | **All** | **P** |
| **ANC** | | | | | | | | | < 0.001*** |
| • <4 (ref.) | 606 (15.07%) | 261 (5.36%) | 120 (9.30%) | 154 (10.82%) | 310 (13.53%) | 248 (24.17%) | 121 (27.94%) | 1820 (11.86%) | |
| • ≥4 | 3416 (84.93%) | 4605 (94.64%) | 1170 (90.70%) | 1269 (89.18%) | 1981 (86.47%) | 778 (75.83%) | 312 (72.06%) | 13531 (88.14%) | |
| **Place of Residence** | | | | | | | | | < 0.001*** |
| • Urban | 1807 (44.93%) | 3312 (68.06%) | 384 (29.77%) | 738 (51.86%) | 841 (36.71%) | 380 (37.04%) | 106 (24.48%) | 7568 (49.30%) | |
| • Rural (ref.) | 2215 (55.07%) | 1554 (31.94%) | 906 (70.23%) | 685 (48.14%) | 1450 (63.29%) | 646 (62.96%) | 327 (75.52%) | 7783 (50.70%) | |
| **Age group of respondents** | | | | | | | | | < 0.001*** |
| •15–19 | 96 (2.39%) | 102 (2.10%) | 32 (2.48%) | 46 (3.23%) | 75 (3.27%) | 51 (4.97%) | 14 (3.23%) | 416 (2.71%) | |
| •20–24 | 544 (13.53%) | 791 (16.26%) | 199 (15.43%) | 231 (16.23%) | 420 (18.33%) | 154 (15.01%) | 75 (17.32%) | 2414 (15.73%) | |
| •25–29 | 1005 (24.99%) | 1222 (25.11%) | 317 (24.57%) | 390 (27.41%) | 555 (24.23%) | 239 (23.29%) | 119 (27.48%) | 3847 (25.06%) | |
| •30–34 | 1146 (28.49%) | 1226 (25.20%) | 328 (25.43%) | 366 (25.72%) | 534 (23.31%) | 260 (25.34%) | 103 (23.79%) | 3963 (25.82%) | |
| •35–39 | 829 (20.61%) | 1021 (20.98%) | 254 (19.69%) | 241 (16.94%) | 429 (18.73%) | 199 (19.40%) | 83 (19.17%) | 3056 (19.91%) | |
| •40–44 | 340 (8.45%) | 419 (8.61%) | 127 (9.84%) | 115 (8.08%) | 233 (10.17%) | 93 (9.06%) | 30 (6.93%) | 1357 (8.84%) | |
| •45–49 (ref.) | 62 (1.54%) | 85 (1.75%) | 33 (2.56%)2 | 34 (2.39%) | 45 (1.96%) | 30 (2.93%) | 9 (2.08%) | 298 (1.94%) | |
| **Have a husband/partner** | | | | | | | | | < 0.001*** |
| • No (ref.) | 126 (3.13%) | 135 (2.77%) | 74 (5.74%) | 44 (3.09%) | 71 (3.10%) | 33 (3.22%) | 25 (5.77%) | 508 (3.31%) | |
| • Yes | 3896 (96.87%) | 4731 (97.23%) | 1216 (94.26%) | 1379 (96.91%) | 2220 (96.90%) | 993 (96.78%) | 408 (94.23%) | 14843 (96.69%) | |
| **Education Level** | | | | | | | | | < 0.001*** |
| • No education (ref.) | 38 (0.94%) | 21 (0.43%) | 56 (4.34%) | 14 (0.98%) | 36 (1.57%) | 8 (0.78%) | 31 (7.16%) | 204 (1.33%) | |
| • Primary | 888 (2.08%) | 1185 (24.35%) | 439 (34.03%) | 404 (28.39%) | 630 (27.50%) | 224 (21.83%) | 89 (20.55%) | 3859 (24.14%) | |
| • Secondary | 2297 (57.11%) | 2979 (61.22%) | 594 (46.05%) | 795 (55.87%) | 1156 (50.46%) | 578 (56.34%) | 229 (52.89%) | 8628 (56.20%) | |
| • Higher | 799 (19.87%) | 681 (14.00%) | 201 (15.58%) | 210 (14.76%) | 469 (20.47%) | 216 (21.05%) | 84 (19.40%) | 2660 (17.33%) | |
| **Parity** | | | | | | | | | < 0.001*** |
| • < 2 | 1161 (28.87%) | 1758 (36.13%) | 370 (28.68%) | 401 (28.18%) | 694 (30.29%) | 267 (26.02%) | 104 (24.02%) | 4755 (30.98%) | |
| • 2–4 | 2572 (63.95%) | 2949 (60.60%) | 757 (58.68%) | 936 (65.78%) | 1362 (59.45%) | 588 (57.31%) | 243 (56.12%) | 9407 (61.28%) | |
| • > 4 (ref.) | 289 (7.19%) | 159 (3.27%) | 163 (12.64%) | 86 (6.04%) | 235 (10.26%) | 171 (16.67%) | 86 (19.86%) | 1189 (7.75%) | |
| **Wealth status** | | | | | | | | | < 0.001*** |
| • Poorest (ref.) | 871 (21.66%) | 474 (9.74%) | 783 (60.70%) | 285 (20.03%) | 858 (37.45%) | 576 (56.14%) | 226 (52.19%) | 4073 (25.53%) | |
| • Poorer | 876 (21.78%) | 805 (16.54%) | 244 (18.91%) | 317 (22.28%) | 517 (22.57%) | 193 (18.81%) | 79 (18.24%) | 3031 (19.74%) | |
| • Midle | 857 (21.31%) | 1050 (21.58%) | 118 (9.15%) | 342 (24.03%) | 351 (15.32%) | 117(11.40%) | 55 (12.70%) | 2890 (18.83%) | |
| • Richer | 752 (18.70%) | 1259 (25.87%) | 77 (5.97%) | 254 (17.85%) | 272 (11.87%) | 103 (10.04%) | 43 (9.93%) | 2760 (17.98%) | |
| • Richest | 666 (16.56%) | 1278 (26.26%) | 68 (5.27%) | 225 (15.81%) | 293 (12.79%) | 37 (3.61%) | 30 (6.93%) | 2597 (16.92%) | |
| **Covered by health insurance** | | | | | | | | | < 0.001*** |
| • No (ref.) | 1455 (36.18%) | 1961 (40.30%) | 499 (38.68%) | 630 (44.27%) | 712 (31.08%) | 482 (46.98%) | 100 (23.09%) | 5839 (38.04%) | |
| • Yes | 2567 (63.82%) | 2905 (59.70%) | 791 (61.32%) | 793 (55.73%) | 1579 (68.92%) | 544 (53.02%) | 333 (76.91%) | 9512 (61.96%) | |

* p < 0.05;
** p < 0.01;
*** p < 0.001.

**Table 2. Binary logistic regression of ANC utilization by regions (n = 15,351).**

| Predictor | ≥4 ANC visits | | | |
|---|---|---|---|---|
| | **P** | **OR** | **Lower Bound** | **Upper Bound** |
| **Regional** | | | | |
| Region: Sumatera | 0.014* | 1.370 | 1.066 | 1.761 |
| Region: Java-Bali | 0.000*** | 3.607 | 2.741 | 4.746 |
| Region: Nusa Tenggara | 0.000*** | 4.365 | 3.229 | 5.899 |
| Region: Kalimantan | 0.000*** | 2.232 | 1.664 | 2.994 |
| Region: Sulawesi | 0.000*** | 1.980 | 1.523 | 2.574 |
| Region: Maluku Islands | 0.171 | 1.213 | 0.920 | 1.600 |

* $p < 0.05$;

** $p < 0.01$;

*** $p < 0.001$.

**Table 3. Binary logistic regression of ANC utilization by socio-demographic (n = 15,351).**

| Predictor | ≥4 ANC visits | | | |
|---|---|---|---|---|
| | **P** | **OR** | **Lower Bound** | **Upper Bound** |
| **Socio-demographic** | | | | |
| Place of Residence: Urban | 0.584 | 0.967 | 0.856 | 1.092 |
| Age group of respondents: 15–19 | < 0.001*** | 0.336 | 0.218 | 0.519 |
| Age group of respondents: 20–24 | 0.038* | 0.675 | 0.465 | 0.979 |
| Age group of respondents: 25–29 | 0.691 | 0.930 | 0.652 | 1.328 |
| Age group of respondents: 30–34 | 0.763 | 1.055 | 0.744 | 1.496 |
| Age group of respondents: 35–39 | 0.441 | 1.145 | 0.811 | 1.618 |
| Age group of respondents: 40–44 | 0.841 | 1.037 | 0.725 | 1.485 |
| Have a husband/partner: Yes | < 0.001*** | 2.107 | 1.674 | 2.651 |
| Education Level: Primary | < 0.001*** | 2.527 | 1.838 | 3.474 |
| Education Level: Secondary | < 0.001*** | 3.882 | 2.815 | 5.353 |
| Education Level: Higher | < 0.001*** | 3.669 | 2.559 | 5.259 |
| Parity: < 2 | < 0.001*** | 3.580 | 2.857 | 4.485 |
| Parity: 2–4 | < 0.001*** | 2.519 | 2.121 | 2.992 |
| Wealth status: Poorer | < 0.001*** | 1.674 | 1.449 | 1.933 |
| Wealth status: Midle | < 0.001*** | 2.056 | 1.739 | 2.431 |
| Wealth status: Richer | < 0.001*** | 2.690 | 2.204 | 3.284 |
| Wealth status: Richest | < 0.001*** | 3.596 | 2.813 | 4.596 |
| Covered by health insurance: Yes | < 0.001*** | 1.485 | 1.334 | 1.653 |

* $p < 0.05$;

** $p < 0.01$;

*** $p < 0.001$.

Table 2 shows the results of the binary logistic regression test, which shows disparities between the regions in the use of ANC in Indonesia. At this stage, <4 ANC visits during pregnancy was used as a reference. Table 3 reveals that all regions show a difference compared to the Papua region as a reference, except the Maluku region, which is not significant and shows differences in the use of ANC compared to the Papua region.

Table 2 shows that the largest difference in the utilization of ≥4 ANC visits is between the Nusa Tenggara and Papua regions. Women in the Nusa Tenggara region have 4.365 times

more than ≥4 ANC visits compared to women in the Papua region (OR 4.365; 95% CI 3.229–5.899). Women in the Java-Bali region were 3.607 times more likely to make ≥4 ANC visits than women in the Papua region (OR 3.607; 95% CI 2.741–4.746).

Table 2 also shows disparities between the Sumatra, Kalimantan and Sulawesi regions compared to the Papua region. Women in the Sumatra region have 1.370 times the chance of making ≥4 ANC visits compared to women in the Papua region (OR 1.370; 95% CI 1.066–1.761). Women in the Kalimantan region had 2.232 times the chance of making ≥4 ANC visits compared to women in the Papua region (OR 2.232; 95% CI 1.664–2.994). Women in the Sulawesi region had 1,980 times the chance of making ≥4 ANC visits compared to women in the Papua region (OR 1.980; 95% CI 1.523–2.574).

In addition to the region category, other variables found to contribute to the predictor are age group, husband/partner, education level, parity, wealth status, and health insurance. Table 3 shows that women in the age group of 15–19 years had a 0.336 times higher chance of making ≥4 ANC visits compared to women in the age group of 45–49 years (OR 0.336; 95% CI 0.218–0.519). The age group of 20–24 years had 0.675 times the chance of making ≥4 ANC visits compared to women in the age group of 45–49 years (OR 0.675; 95% CI 0.465–0.979). This shows that the youngest age group has a lower possibility of ≥4 ANC visits than the oldest age group.

Table 3 indicates that women who have a husband/partner have a better chance of making ≥4 ANC visits than those without a husband/partner. More specifically, women who have a husband/partner have a 2.107 times higher chance of making ≥4 ANC visits compared to women who do not have a husband/partner (OR 2.107; 95% CI 1.674–2.651).

Table 3 shows that women with higher levels of education have a better chance of making ≥4 ANC visits than those without higher levels of education. Women with primary education had 2.527 times the chance of making ≥4 ANC visits compared to women with no education (OR 2.527; 95% CI 1.838–3.474). Women with secondary education were 3.882 times more likely to make ≥4 ANC visits compared to women with no education (OR 3.882; 95% CI 2.815–5.353). Women with a higher level of education had 3.669 times the chance of making ≥4 ANC visits than women with no education (OR 3.669; 95% CI 2.559–5.259).

Table 3 shows that women with lower parity have a better chance of making ≥4 ANC visits than those who have a parity >4. Women who have a parity <2 have 3.580 times the chance of making ≥4 ANC visits than women who have a parity >4 (OR 3.580; 95% CI 2.857–4.485). Women who had parity between 2 and 4 had 2.519 times the chance of making ≥4 ANC visits compared to women who had a parity >4 (OR 2.519; 95% CI 2.121–2.992).

Table 3 shows that the higher the wealth status held by a woman, the higher the probability of making ≥4 ANC visits. The richest women had 3.596 times the chance of making ≥4 ANC visits compared to the poorest women (OR 3.596; 95% CI 2.813–4.586).

Table 3 shows that women covered by health insurance had a better chance of making ≥4 ANC visits than those who were not covered. Women who are covered by health insurance have 1.485 times the chance of making ≥4 ANC visits compared to women who are not covered by health insurance (OR 1.485; 95% CI 1.334–1.653).

## Discussion

The results showed that disparity between regions in the use of ANC is still ongoing. The disparity is also clearly seen between the eastern and western regions. The results of this analysis are in line with several studies in Indonesia that show that the eastern region lags behind the western region [19, 20], especially when compared to the Java-Bali region as the center of government.

Geographically, conditions in eastern Indonesia also show more extreme variability than conditions in the western regions. These conditions make some parts of eastern Indonesia fall in the category of an isolated or remote area [20–22], and some other areas are quite difficult to reach because of the limited means of available roads and public transportation [19].

Qualitatively, some research also shows that in the eastern region, having more health beliefs is a challenge for health workers to strive for better maternal health [23, 24]; this not only applies to the community but also applies to the health belief encompassed by health workers because they are an inseparable part of the community itself [25].

The analysis shows that there is no difference between urban and rural areas in ANC utilization in Indonesia. This condition is different from the findings in Nigeria [26], Ethiopia [9], Pakistan [27] and several other countries [28], which found disparities between urban and rural areas.

The age group was found to be a predictor of ANC utilization. The youngest age group has a lower probability of making ≥4 ANC visits. This is likely due to a lack of experience, so knowledge about health risks is lower [29, 30]. A study in India that analyzed the relationship between child marriage and the utilization of maternal healthcare services concluded that many challenges were found; more effort was needed so that child marriage could have a positive impact on the use of maternal healthcare services [31].

The analysis shows that women who have husbands/partners are more likely to use ANC. This is in line with the findings of several studies that have shown the role of a husband/partner in providing support for a woman's healthy lifestyle [32–35]. Some other studies actually encourage a husband to help improve a woman's health status by actively encouraging a healthier lifestyle [36, 37].

The analysis of this study proves that education is one of the determining factors for women in Indonesia to make ≥4 ANC visits. In general, it can be explained that the more educated a person is, the easier it is to receive new health information and understand the dangers or risks of behaviors that have an impact on health [38–40]. Education has also been shown to play a role in one's perception of the quality of health services [41, 42]. Furthermore, improving education is generally accepted as one of the determinants of life expectancy [43].

This study found that parity is a determinant of the use of ANC. The lower the parity, the more likely one is to make ≥4 ANC visits. Parity as one of the determinants of ANC utilization has also been found in several recent studies in several countries [44–46].

In line with the level of education, wealth status was also found to be directly proportional to the likelihood of ≥4 ANC visits. This result is in accordance with several studies that found that wealth status is one of the positive determinants of ANC utilization, namely, in Ethiopia [47], Pakistan [48], Nigeria [49], and Uganda [50]. The higher the wealth status of a woman is, the more likely the woman is to make ≥4 ANC visits.

Women covered by health insurance were found to have higher ANC utilization. Women who did not have health insurance had lower ANC utilization. This finding is in line with the goal of the National Health Insurance released by the Indonesian government to provide universal access to health care facilities [51, 52]. Social insurance policies to increase public access to health care facilities have also been adopted by other countries. The results of other studies that have evaluated this matter have shown positive results [53–55], although there were still some obstacles encountered in the implementation [56, 57].

The disparities found and detected in this study are still limited to a superficial understanding. Researchers suggest that further research be carried out to detect the in-depth causes of disparity.

## Conclusions

Based on the results of this study, it can be concluded that there were interregional disparities was in ≥4 ANC visits during pregnancy in Indonesia. Besides regional, other influential variables were husband/partner, education level, parity, wealth status, and health insurance.

The structured policy is needed to reach regions that have low coverage of ≥4 ANC. Policymakers need to use the results of this study to take the necessary policies. Policies that focus on service equality to reduce disparities.

## Author Contributions

**Conceptualization:** Agung Dwi Laksono.

**Data curation:** Agung Dwi Laksono, Rukmini Rukmini, Ratna Dwi Wulandari.

**Formal analysis:** Agung Dwi Laksono, Rukmini Rukmini, Ratna Dwi Wulandari.

**Funding acquisition:** Ratna Dwi Wulandari.

**Investigation:** Rukmini Rukmini.

**Methodology:** Agung Dwi Laksono.

**Project administration:** Rukmini Rukmini.

**Resources:** Rukmini Rukmini.

**Software:** Rukmini Rukmini, Ratna Dwi Wulandari.

**Supervision:** Ratna Dwi Wulandari.

**Validation:** Agung Dwi Laksono, Ratna Dwi Wulandari.

**Visualization:** Agung Dwi Laksono.

**Writing – original draft:** Agung Dwi Laksono, Rukmini Rukmini.

**Writing – review & editing:** Ratna Dwi Wulandari.

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
