## [Decision Letter · Decision Letter 0]

18 Nov 2019

PONE-D-19-27564

Regional Disparities in Antenatal Care Utilization in Indonesia

PLOS ONE

Dear Dr. Ratna Dwi,

Thank you for submitting your manuscript to PLOS ONE. After careful consideration, we feel that it has merit but does not fully meet PLOS ONE’s publication criteria as it currently stands. Therefore, we invite you to submit a revised version of the manuscript that addresses the points raised during the review process.

I would like to applause the authors for taking this initiative to research the regional disparity of Antenatal Care Utilization in the study area. As it has been indicated in this study ANC is the best strategy to uphold the well-being of the mothers, the unborn baby and to perpetuate healthy and productive generation.Being said that, the following are point by point comments need further revision.Abstract:This section precisely illustrate and expound the entire study. But there is enumeration discrepancy in the result section. Perhaps, this would be due to, I believe, an honest typing errors. In fact, it would be good to have the confidence interval in each regression analysis to demonstrate the estimate computed from the statistics of the observed data and to clearly show where the estimate laid. On the other hand, the main purpose of any research is to identify the pre-existing gap or problem and recommend based on the study finding. It appears to be there is no recommendation incorporated in the section. It would be great to include a brief recommendation under conclusion.Introduction:In general, this section encompass necessary facts that provide important information related to ANC service in various regions in Indonesia. However, it didn’t include literature of similar study from different countries with the same setting. Having those literature will help to visualize the gap existed in this study area and in order to draw the right argument in the discussion section.Methods:• What was your inclusion and exclusion criteria to select the variables from IDHS?• What was your operation definition for your dependent variable (ANC utilization)?• You need to clearly depict the methods or criteria employed in this study to interpret and identify regional discrepancy.• It would be very important to have a brief description related to other important variables that could have direct or indirect impact in this the study.Result:This section need further work up. Here are some of my observation you need to pay attention to.• What is your rationale to merge regression analysis with socio-demographic characteristics? • Make sure you separate the socio-demographic characteristics with those sub titles in this section. • Make sure you address the objective clearly.• I assume ANC utilization is considered as a dependent variable and more than two independent variable have been included as well in this study, then why your analysis clogged on binary logistic regression. Don’t you think additional analysis would help to refine your result? • There is inconsistency throughout this section.• I recommend to conduct multiple logistic regression analysis to provide concrete result to assert the disparity. Other than this, I don’t see any issue in the write up in this section, but as I indicated above the analysis appears to be incomplete. In General, the result section could impact the discussion and conclusion section.Discussion:Well written with clear and evidence based argument. However, this would be an overdue until complete analysis conducted. 

Even though, the conclusion section looks well written, I assume you will further rewrite after reanalysis.

We would appreciate receiving your revised manuscript by Jan 02 2020 11:59PM. To enhance the reproducibility of your results, we recommend that if applicable you deposit your laboratory protocols in protocols.io, where a protocol can be assigned its own identifier (DOI) such that it can be cited independently in the future. For instructions see: http://journals.plos.org/plosone/s/submission-guidelines#loc-laboratory-protocols

We look forward to receiving your revised manuscript.

Kind regards,

Solomon Assefa Woreta

Academic Editor

PLOS ONE

Journal Requirements:

2. Please correct your reference to "p=0.000" to "p<0.001" or as similarly appropriate, as p values cannot equal zero.

Additional Editor Comments (if provided):

I would like to applause the authors for taking this initiative to research the regional disparity of Antenatal Care Utilization in the study area. As it has been indicated in this study ANC is the best strategy to uphold the well-being of the mothers, the unborn baby and to perpetuate healthy and productive generation.

Being said that, the following are point by point comments need further revision.

Abstract:

This section precisely illustrate and expound the entire study. But there is enumeration discrepancy in the result section. Perhaps, this would be due to, I believe, an honest typing errors. In fact, it would be good to have the confidence interval in each regression analysis to demonstrate the estimate computed from the statistics of the observed data and to clearly show where the estimate laid. On the other hand, the main purpose of any research is to identify the pre-existing gap or problem and recommend based on the study finding. It appears to be there is no recommendation incorporated in the section. It would be great to include a brief recommendation under conclusion.

Introduction:

In general, this section encompass necessary facts that provide important information related to ANC service in various regions in Indonesia. However, it didn’t include literature of similar study from different countries with the same setting. Having those literature will help to visualize the gap existed in this study area and in order to draw the right argument in the discussion section.

Methods:

• What was your inclusion and exclusion criteria to select the variables from IDHS?

• What was your operation definition for your dependent variable (ANC utilization)?

• You need to clearly depict the methods or criteria employed in this study to interpret and identify regional discrepancy.

• It would be very important to have a brief description related to other important variables that could have direct or indirect impact in this the study.

Result:

This section need further work up. Here are some of my observation you need to pay attention to.

• What is your rationale to merge regression analysis with socio-demographic characteristics?

• Make sure you separate the socio-demographic characteristics with those sub titles in this section.

• Make sure you address the objective clearly.

• I assume ANC utilization is considered as a dependent variable and more than two independent variable have been included as well in this study, then why your analysis clogged on binary logistic regression. Don’t you think additional analysis would help to refine your result?

• There is inconsistency throughout this section.

• I recommend to conduct multiple logistic regression analysis to provide concrete result to assert the disparity.

Other than this, I don’t see any issue in the write up in this section, but as I indicated above the analysis appears to be incomplete.

In General, the result section could impact the discussion and conclusion section.

Discussion:

Well written with clear and evidence based argument. However, this would be an overdue until complete analysis conducted.

Reviewers' comments:

Reviewer's Responses to Questions

**Comments to the Author**

1. Is the manuscript technically sound, and do the data support the conclusions?

Reviewer #1: Partly

Reviewer #2: Yes

2. Has the statistical analysis been performed appropriately and rigorously? 

Reviewer #1: Yes

Reviewer #2: Yes

3. Have the authors made all data underlying the findings in their manuscript fully available?

Reviewer #1: Yes

Reviewer #2: Yes

4. Is the manuscript presented in an intelligible fashion and written in standard English?

Reviewer #1: Yes

Reviewer #2: Yes

5. Review Comments to the Author

Reviewer #1: Overall:

Although this study does not demonstrate the advance in the field, the manuscript has been identified as original study with 16% similarity index. The authors have used large and sufficient amount of secondary data that can be accessed online but with registration and permission only through the link: https://dhsprogram.com/data/available-datasets.cfm. Overall, the authors have summarized the main research question and key findings. Authors have also identified other literature on the topic and explain how the study relates to this previously published research. However, the authors should put more explanation on the rationale and significance of this study, particularly in the introduction. The figures and tables are clear, readable, and support the findings. There are only some captions and labels need further clarifications. The presentation of figures and tables are appropriate for the type of data being presented. There is no experiments or interventions used in this study since the authors collected the data from the 2017 Indonesian Demographic Data Survey (IDHS) that can be access online subject to registration and permission to use the data. The authors have used enough qualitative data to draw a conclusion and addressed possible limitations of the research. The process of collecting data, selecting variables of the study, and analyzing the data still needs further details to allow other researchers to fully replicate or recreate the analysis and validate the study. The authors have followed best practices for reporting and conformed to ethical guidelines. The authors have used one or more of the highly qualified native English speaking editors at American Journal Experts (AJE) to edit the manuscript for proper English language, grammar, punctuation, spelling, and overall style. The results of this study support the conclusions even though some of them could not be justified directly from the results. The authors briefly mentioned about the limitation of this study. The statistical analysis was adequate but needs more details on the assumption that we need to meet for each single type of statistical test to ensure the validity of the results. The summary data presented in the manuscript have provided enough evidence for the author’s conclusions although the necessary data points can only be accessed online with registration and permission.

Below is given point-by-point comments to the manuscript:

Title:

The title has been clear and concise but there is inconsistency in writing the title between in the cover page (Regional Disparities in Antenatal Care Utilization in Indonesia) and the first page of the original manuscript (Regional Disparities of Antenatal Care Utilization in Indonesia).

Abstract

The abstract has mentioned the main objective of the study, explained how the study was done, summarized some important results but less explanation on their rationale and significance.

Introduction:

In the last sentence: This study aims to analyze interregional disparities in ANC utilization with ≥ 4 visits during pregnancy in Indonesia.

Methods:

This research has used secondary data derived from IDHS 2017. The data are accessible online through https://dhsprogram.com/data/available-datasets.cfm but with registration and permission.

You have missed to mention about the use of principal component analysis and chi-square in your analysis here.

Results:

1st sentence: …………………….in the use of ANC with ≥ 4 visits compared to………..

Conclusion:

…………. in ANC utilization with ≥ 4 visits between …..

There is inconsistency in the number of variables cause disparity in the ANC utilization ≥ 4 visits in Indonesia mentioned in the conclusion (abstract) (6) and the conclusion (main manuscript) (10).

Keywords:

Antenatal care utilization, contributing factors, regional disparity, Indonesia.

Main Manuscript

Introduction

The introduction does not provide sufficient background that puts the manuscript into context. This is because:

1. Some information between paragraphs is not well-linkage so that it is difficult to understand the rationale, purpose, and significance of the study.

2. Lack of data and review of key literature regarding disparity in the ANC utilization to show what the problem with disparity, why it is important to be addressed, and controversies or disagreements in the field.

These result in unclear statement of the overall aim and significance of the study.

Below some disconnections found in every paragraph in the Introduction:

1st paragraph:

You can simply said:

“In the 2015-2019 National Medium-Term Development Plan, Indonesia has targeted to increase the distribution equity and quality of health services by the end of 2019”. Then you can combined and continue this sentence with the second paragraph like: “The target can be determined by three indicators, namely …..”.

2nd paragraph:

You mentioned about three indictors to increase the distribution equity and quality of health services and there is no linkage information to the use of ANC with ≥ 4 visits.

3rd paragraph:

You mentioned about target of mortality rates with no further linkage to the effort of increasing the distribution equity and quality of health services.

4th paragraph:

You mentioned about disparities in maternal deaths among districts/cities in Indonesia which the highest is in Eastern Indonesia and the risk factors that most influenced maternal mortality, including low coverage of four pregnancy visits. This is good information and try to connect it with what you have written in the 5th paragraph.

5th paragraph:

You have introduced ANC as the main strategy for reducing maternal mortality and morbidity but please check the structure of the 1st sentence. I would recommend if you can write it as “Antenatal care (ANC) is the main strategy to decrease maternal morbidity and mortality”.

You also mentioned a good information about the increase use of ANC and its variation (disparity) among populations groups due to geographical, demographic, socioeconomic, and cultural differences that result in the decrease of access to service as well as its quality and affordability.

6th paragraph:

You mentioned some statistics showing the increase proportion of ANC utilization (1st and 4th visits) and identified that the quality of ANC services needs further improvement. After that, however, you put some information that does not link with what you have said in the previous 1st and 2nd lines. The information perhaps can be used in Discussion rather than in the Introduction.

7th paragraph:

The aim of study was to analyze disparity in the utilization of ANC with ≥ 4 visits to provide clear directions for the Health Ministry to complete regional priority data in an effort to reduce maternal mortality. However, it is not clear enough on how you could provide directions to the Ministry, what regional priority data is for and the connection with the reduction of maternal mortality. Please further explain them in the discussion so that you can answer your research question.

You have mentioned in the 6th paragraph that the quality of ANC services needs to be improved. How this relates to the utilization of ANC with ≥ 4 visits?

Methods

Data Source

Since you are using secondary data from IDHS which can be accessed online with registration/permission, there is no need to explain too much detail on the sampling design of the survey, except you did the “real” survey. The sampling design has been perhaps explained in details in reference no 13, so you do not need to repeat that again here. In this part, you should provide enough detail on how (the procedures) you get the data from IDHS, what variables that you use (dependent and independent variables), how to select them, and how many data you get from the IDHS to answer your research question to allow suitably skilled investigators to fully replicate your study. Please mention here that you have obtained the permission to use the data for the purpose of the study.

Procedure

Since you are using secondary data from IDHS and not conducting the “real” survey, there is no need to explain the ethical clearance of conducting the survey.

I do not think this sub section “Procedure” is needed since you are not dealing with primary data or experimental study.

Data Analysis

The explanation about variables used in the study can be placed under “Data Source” as I mentioned earlier.

In this section, you should you provide sufficient information on how you analyze the data. This includes how the principal component analysis was used to calculate the score; how to arrange the national wealth quintiles; how to select variables that related to the frequency of ANC utilization using the chi-square test, including the assumption that we need to meet; and how to determine disparity using binary logistic regression to allow suitably skilled investigators to fully replicate your study.

I could not see the use of chi-square test and principal component analysis in your results.

Results

1st paragraph:

Fig. 1 caption: “Distribution of ANC utilization percentage with ≥ 4 visits across 34 provinces in Indonesia”.

Legend: “Antenatal care with ≥ 4 visits”

I would suggest that the 1st sentence will be written as following:

“According to Fig. 1, the eastern part of Indonesia (Maluku and Papua regions) has the lowest percentage of ANC utilization with ≥ 4 visits (< 75.64%). This was higher in the western part of Indonesia (Sumatera), recorded from 75.65% to 92.74% and best centered in the central part of Indonesia (Java-Bali), > 92.74%”.

2nd paragraph:

The first sentence: The statistical description, calculated in counts (%), of female ……

The second sentence: Table 1 shows that there are statistically significant socio-demographic differences between regions. Also, please mention here or in the “Methods” which category of variables become references.

You should also explain why the 34 provinces are divided into 7 regions in the “Methods”.

Table 1:

In the 9th column, please change the label “All” to “Total”.

In the last column, please put “p-value” instead of “P” which in Statistics can be meant proportion. Also, provide information of what test statistics used for testing the differences among regions and the testing criteria in “Methods”.

3rd and 4th paragraphs:

Please combine them into one paragraph.

5th paragraph:

The third sentence: Typo “Table 3” should be “Table 2”; “………………….and shows no differences……………”

6th paragraph:

The first sentence: “………………….differences in the ANC utilization with ≥ 4 visits is …….”

The second sentence: “……..Nusa Tenggara region were 4.365 times more likely to have ≥ 4 ANC visits compared…..”

7th paragraph:

Combine this paragraph with the 6th paragraph.

8th paragraph:

Please be aware of the use of articles “a”.

The fourth sentence: “…………………has a lower tendency to utilize ANC ≥ 4 visits than…..”.

Table 2: I suggest the label for column “≥ 4 ANC visits” to be “ANC utilization with ≥ 4 visits”.

9th paragraph:

Please combine the 1st and 2nd sentences together, so that “Table 2 indicates that women who have a husband/partner have 2.107 times higher chance of utilizing ANC ≥ 4 visits compared to women who do not have a husband/partner (OR 2.107; 95%CI 1.674-2.651)”.

Please be aware of the use of articles “a” in the 9th and 11th paragraphs.

Please combine the 9th to the 13th paragraph into one concise summary.

Discussion

Please combine some of paragraphs that contain less than 3 sentences or add more sentences to make a proper paragraph.

Please explain the 3rd paragraph using simpler language.

Conclusion

Here, you have mentioned 10 variables that contributed to disparities in ANC utilization among women in Indonesia to make ≥ 4 visits. This number is inconsistent with what you have written in the conclusion of the abstract. Some of the variables, such as “not being able to read”, “not being exposed to the media”, “never using the Internet”, “not knowing the signs of danger related to pregnancy”, and “a belief in traditional birth attendants”, may need more explanation and justification in the results to avoid overreach conclusion.

Reviewer #2: Reviewer’s report

Title: Regional Disparities in Antenatal Care Utilization in Indonesia

Version: 1 Date: 1 Nov 2019

Reviewer: I Wayan Gede Artawan Eka Putra

Abstract:

The results need to shorten and use effective sentence.

Please provide also the main implication of this study in the conclusion

Background section:

Need further argumentation about the using of category ≥4 ANC and why is it important?

Methods section is well written.

Results section:

Table 1&2: If p value on the analysis results is 0.000 please write <0.001 on the table.

If p value had been written in the table than the foot note is not necessary.

Table 2: Please provide the reference category of each variable so will be easier to interpret the table.

The age categorization need to be simpler for example divided into 3 categories <20, 20-34, and ≥35 years old.

Discussion Section:

Need further discussion regarding the implication of main result to improve antenatal care utilization in Indonesia. This discussion will be a guidance to write specific recommendations.

The second and third paragraph in discussion section may be merged into one paragraph.

Conclusion:

The recommendation need to be specify, therefor the discussion regarding the implications of main results were important to build a specific recommendation. This will be the main massage of this study.

Declaration of competing interests: I declare that I have no competing interests.

6. PLOS authors have the option to publish the peer review history of their article (what does this mean?). If published, this will include your full peer review and any attached files.

Reviewer #1: No

Reviewer #2: Yes: I Wayan Gede Artawan Eka Putra

---

## [Author Response · Author response to Decision Letter 0]

19 Dec 2019

Reviewer 1: I have incorporated all of your suggestions into my revisions. They were very helpful. Thank you. I included Socio-demographic analysis in binary logistic regression. This analysis will provide clear guidance for policymakers of clear targets.

Reviewer 2: I have incorporated all of your suggestions into my revisions. They were very helpful. Thank you

---

## [Editor Report · Decision Letter 1]

13 Jan 2020

Regional Disparities in Antenatal Care Utilization in Indonesia

PONE-D-19-27564R1

Dear Dr. Ratna Dwi Wulandari,

We are pleased to inform you that your manuscript has been judged scientifically suitable for publication and will be formally accepted for publication once it complies with all outstanding technical requirements.

With kind regards,

Solomon Assefa Woreta

Academic Editor

PLOS ONE
---

## [Editor Report · Acceptance letter]

15 Jan 2020

PONE-D-19-27564R1 

Regional Disparities in Antenatal Care Utilization in Indonesia 

Dear Dr. Wulandari:

I am pleased to inform you that your manuscript has been deemed suitable for publication in PLOS ONE. Congratulations! Your manuscript is now with our production department. 

With kind regards,

on behalf of

Dr. Solomon Assefa Woreta 

Academic Editor

PLOS ONE